# Late Cenozoic exhumation in the eastern Junggar Basin: Evidence from zircon (U-Th)/He ages of combustion metamorphic rocks

**Bin Chen**[1,2]*, **Pan Liu**[1], **Yan Dong**[3], **Changjuan Feng**[1], **Zhuang Zhao**[1], **Chaoqun Yang**[4], **Yixin Dong**[4]

1 School of Geography and Environment, Liaocheng University, Liaocheng, China, 2 Key Laboratory of Deep-time Geography and Environment Reconstruction and Applications of Ministry of Natural Resources, Chengdu University of Technology, Chengdu, China, 3 Institute of Sedimentary Geology, Chengdu University of Technology, Chengdu, China, 4 State Key Laboratory of Oil and Gas Reservoir Geology and Exploitation, Chengdu University of Technology, Chengdu, China

* chenbin@lcu.edu.cn

## Abstract

Since the Cenozoic, the peripheral orogenic belts around the Junggar Basin have undergone substantia uplift in response to far-field deformation associated with the India–Asia collision. However, conventional geochronological methods commonly provide only indirect or insufficiently resolved constraints on the timing and geomorphic expression of late Cenozoic uplift and exhumation. Combustion metamorphic (CM) rocks, generated when coal seams are brought into shallow, oxygen-rich conditions during tectonic uplift and denudation and subsequently ignite spontaneously, offer a potential near-surface chronometer for these processes. In this study, we characterized coal maceral composition, rank, and spontaneous combustion tendency; documented the distribution, petrography, and mineral assemblages of CM rocks through field investigations, thin-section observation, and X-ray diffraction; and applied zircon (U-Th)/He dating to constrain the timing of CM rock formation. Three zircon grains define a tightly clustered Middle Pleistocene population with a weighted mean age of 0.63 ± 0.19 Ma, which we interpret as the principal timing of coal seam combustion and CM rock formation. Two older single-grain ages (8.7 ± 0.5 Ma and 87.7 ± 5.4 Ma) are treated cautiously as incompletely reset or inherited pre-combustion thermochronologic components rather than as independent combustion events. The spatial distribution and ages of CM rocks show a clear correspondence with late Cenozoic uplift and exhumation of the orogenic belt. These results demonstrate that zircon (U-Th)/He thermochronology of CM rocks can provide a useful chronological marker for near-surface tectonic processes and offers an additional approach for reconstructing late Cenozoic tectonic evolution in intracontinental orogenic settings.

**Data availability statement:** All relevant data are within the manuscript and its Supporting Information files.

**Funding:** This research was funded by the National Natural Science Foundation of China (grant no. 42201006) and by the Open Fund (DGERA 20221105) of the Key Laboratory of Deep-time Geography and Environment Reconstruction and Applications of Ministry of Natural Resources, Chengdu University of Technology.

**Competing interests:** The authors have declared that no competing interests exist. All authors confirm that there are no competing interests, financial or otherwise, related to this work.

# 1 Introduction

The peripheral orogenic belts of the Junggar Basin represent one of the most active intracontinental orogenic systems, and studies on its tectonic evolution and formation mechanisms are of great significance for basin–mountain coupling, as well as for petroleum and sandstone-type uranium exploration [1–6]. Over the past decades, numerous studies have investigated the tectonic evolution of these orogenic belts from various perspectives [7–11]. Nevertheless, accurately constraining the timing of tectonic evolution remains both a central focus and a major challenge in regional tectonic research.

The Junggar Basin has been successively modified by episodic Mesozoic–Cenozoic tectonic events, resulting in uplift and exhumation along the basin margins and folding or tilting within the basin interior [12–16]. These tectonic events have been primarily identified through stratigraphic contact relationships or thermochronological analyses of magmatic rocks, including apatite fission-track dating and thermal history modeling [8,17–19]. Apatite fission track data have revealed four major episodes of uplift and erosion during the Middle–Late Jurassic, Cretaceous, Paleogene, and Miocene to Present, respectively [20–24]. However, due to the complex Cenozoic tectonic evolution of the Junggar Basin, the limited extent of regional uplift and surface erosion, and the variability in datasets and analytical approaches, a consensus on the subdivision of tectonic events has not yet been reached. Furthermore, the timing of tectonic uplift remains poorly constrained, particularly because direct geochronological evidence for rapid uplift events since the Miocene is relatively scarce [14,25,26]. Combustion metamorphic (CM) rocks, formed by the spontaneous combustion of coal seams following its uplift to near-surface conditions during orogenesis, record formation ages that to some extent reflect the tectonic uplift process.

The formation of CM rocks is closely associated with tectonic uplift and exhumation. CM rocks are primarily distributed along the frontal margins of orogenic belts and occur sporadically at the junction between the Junggar Basin and its surrounding orogenic belts [27–29]. During tectonic uplift, coal seams within strata are exposed to near-surface, oxygen-rich conditions, triggering spontaneous combustion and causing high-temperature metamorphism of the country rocks, forming CM rocks [30–33]. As the main products of coal spontaneous combustion, the formation ages of CM rocks record the timing of coal seam combustion [34,35]. Because coal becomes less prone to self-ignition after prolonged weathering, coal seam combustion generally occurs shortly after uplift and exposure, so that the ages of CM rocks can represent the timing of uplift and erosional exposure of their host strata [34,36–38]. As tectonic uplift progresses, coal seams are sequentially exposed from top to bottom and combust at near-surface conditions, forming CM rocks at different elevations along the frontal margin of the orogenic belt [39]. The ages of CM rocks record the frequency of coal seam combustion and also reflect the rapid tectonic uplift experienced since coal deposition [37–40].

The high temperatures generated during coal seam spontaneous combustion can exceed 1000 °C [41,42]. In some natural coal-fire systems outside the study area, such heating has been inferred or documented to persist for hundreds of years

[43,44]. However, in the eastern Junggar Basin, the available evidence does not directly constrain the duration of individual coal-fire events; rather, it constrains the timing of CM rock formation. Such thermal events are capable of fully or partially resetting the apatite and zircon (U-Th)/He systems in CM rocks, given the low closure temperatures of apatite (~70 °C) and zircon (~180–200 °C) [45–47]. Experimental and empirical studies on the thermal sensitivity of apatite and zircon indicate that short-lived high-temperature heating events (>600 °C), such as wildfires, can result in >90% helium loss in apatite grains and approximately 15% helium loss in zircon grains [48]. Furthermore, Reiners (2005) demonstrated that under relatively short-duration, high-temperature conditions, the kinetics of fission-track annealing in zircon are faster than those of helium diffusion, leading to fission-track ages that are younger than corresponding (U-Th)/He ages [45]. This relationship is widely regarded as a diagnostic indicator of transient reheating events. In addition, zircon (U-Th)/He dating of CM rocks has been successfully applied in the Powder River Basin of Wyoming and Montana [35,44,49]. Consequently, zircon (U-Th)/He ages of CM rocks can provide direct chronological constraints on coal seam combustion and related near-surface uplift-exhumation processes.

Based on detailed studies of the spatial distribution and ages of CM rocks, the timing and rate of tectonic uplift can be constrained, while paleoenvironmental and paleoclimatic information recorded by CM rocks can also be obtained, providing a valuable framework and more comprehensive insights into the Cenozoic tectonic evolution of peripheral orogenic belts [27,35,36,44]. Although extensive research has been conducted on the petrological and geochemical characteristics of CM rocks worldwide, studies focusing on their geochronology and constraints on tectonic uplift remain relatively scarce, and are mainly concentrated in the United States and Central Asia [34,35,37,39]. CM rocks distributed along the frontal margin of the peripheral Junggar orogenic belts provide an ideal setting to investigate the role of these rocks in constraining tectonic evolution since the Miocene [27–29,50].

In this study, integrated field investigations and sampling, combined with previously published data, are employed to characterize the spatial distribution, petrology, and mineralogy of CM rocks using thin-section and X-ray diffraction analyses. Zircon (U-Th)/He dating is systematically applied to determine the formation ages of these rocks. Based on the obtained age data, we aim to constrain the timing of tectonic uplift, clarify the tectonic evolution and controlling factors in the study area, and provide new data and approaches for investigating Cenozoic tectonic evolution since the Miocene.

## 2 Geological setting

The eastern uplift of the Junggar Basin is bounded by the Kelameili Mountains to the north and the Bogda Mountains to the south (Fig 1a) and comprises 15 secondary structural units, including the Wucaiwan Depression, the Shazhang Fold-Thrust Belt, the Shishugou Depression, and the Shaqi Uplift (Fig 1b) [3,14,15,21,51]. Since the Mesozoic, this region has undergone multiple tectonic events, resulting in four major phases of uplift and denudation: Late Triassic–Early Jurassic compressional uplift, Late Jurassic–Early Cretaceous compressional uplift, Late Cretaceous–Paleocene uplift and tilting, and continuous uplift and erosion from the Miocene to the present [52–56].

The study area is situated within the Shazhang Fold-Thrust Belt, bounded to the north by the Kelameili Mountains and to the south by the Shaqi Uplift. From west to east, Shazhang Fold-Thrust Belt consists of five tertiary structural units: the Shaqiuhe Anticline, the Lucaogou Syncline, the Huoshaoshan Anticline, the Xidagou Syncline, and the Zhangpenggou Anticline (Fig 1c, d). Several high-angle thrust faults are developed in the area, including the Shaxi Fault, the Huodong Fault, and the Zhangdong Fault, which initiated activity in the Early Permian and were finally established by the end of the Cretaceous [3,4,52].

The southern sector of the study area is mantled by Quaternary deposits, while the central part exhibits a narrow, belt-shaped outcrop zone composed of the Cretaceous Tugulu Group, the Jurassic Shishugou Formation ($J_{2-3}$sh), Xishanyao Formation ($J_2$x), Sangonghe Formation ($J_1$s), Badaowan Formation ($J_1$b), and the Triassic Xiaoquangou Group [57–59]. Multiple phases of tectonic activity have resulted in four regional unconformities: between the Jurassic Badaowan Formation and the Triassic Xiaoquangou Group ($J_1$b/$T_{2-3}$xq), between the Cretaceous Tugulu Group and the Jurassic Shishugou

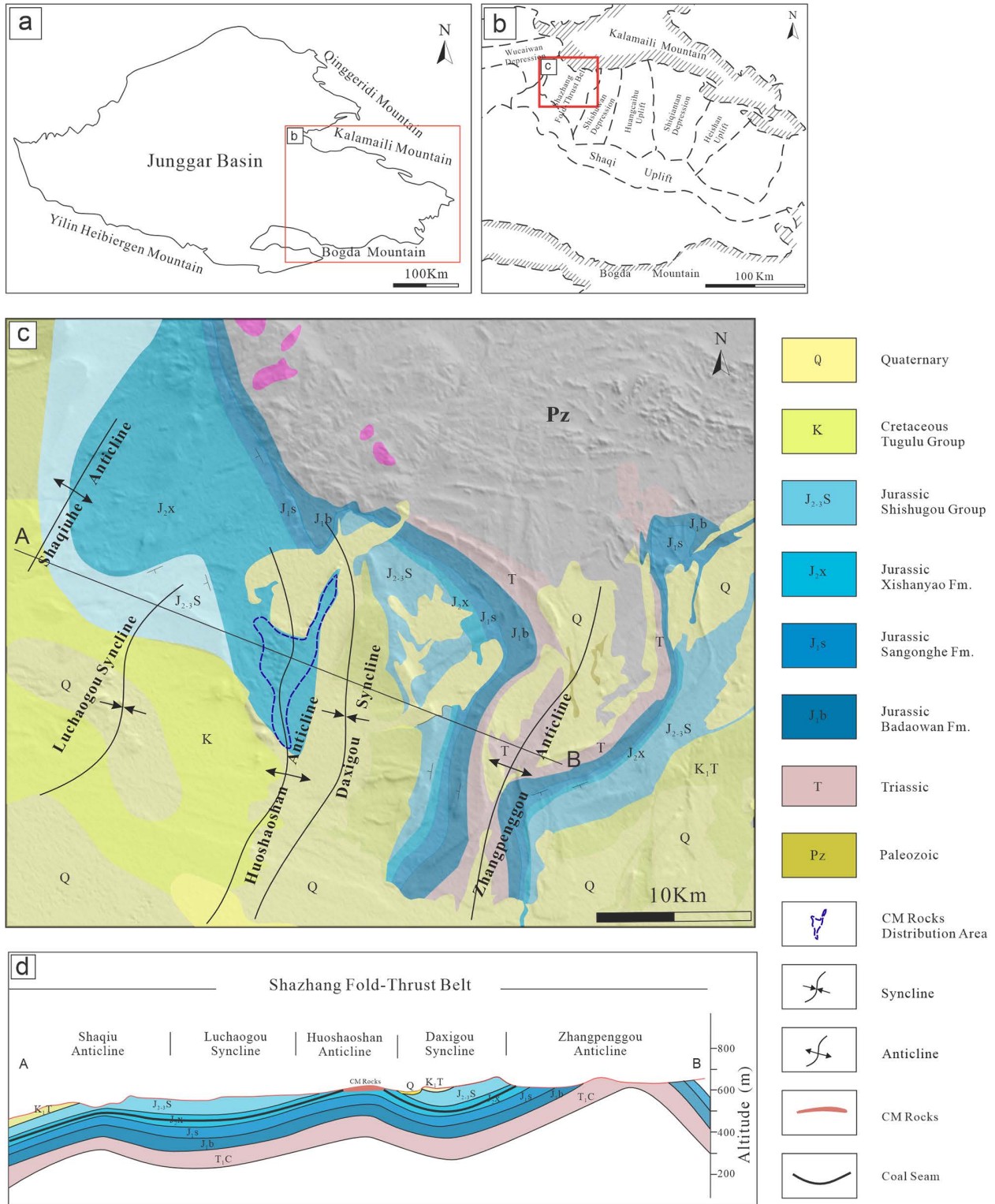

**Fig 1. (a) Regional location of the Junggar Basin.** (b) Simplified structural framework of the eastern Junggar Basin. (c) Simplified geological sketch of the study area, showing the distribution of CM rocks. The DEM base was derived from NASA Earth Observatory data. (d) Schematic E–W cross-section of the Shazhang Fold-Thrust Belt.

Formation ($K_1tg/J_{2-3}sh$), between the Neogene Shawan Formation and the Cretaceous Tugulu Group ($N_1s/K_1tg$), and between the Quaternary deposits and the Neogene Shawan Formation ($Q/N_1s$) [4,12] (Fig 2).

Coal-bearing strata in the area include the Middle–Upper Jurassic Shishugou Formation Lower Submember ($J_{2-3}s$), the Xishanyao Formation ($J_2x$), and the Badaowan Formation ($J_1b$). The Xishanyao Formation constitutes the principal coal-bearing unit, hosting three relatively stable seams ($B_2$, $B_1$ and $B_0$). Among them, B1 is the thickest (average 26 m), whereas $B_0$ and $B_2$ average 2 m and 14 m, respectively [60,61]. Overall, the coal seams of the Xishanyao Formation are stratigraphically stable, show limited thickness variation, and are relatively shallowly buried. In the central part of the area, uplift related to the Huoshaoshan Anticline caused thinning or local pinching-out of the seams, whereas seam thickness increases toward both limbs of the anticline. Near the axis of the Xidagou Syncline, coal seams reach their maximum thickness and then taper gradually toward the flanks. Along the north–south direction, seam thickness increases in the central sector but decreases toward both margins, with local depositional absence in the northern and southern sectors [62,63]. The Lower Submember of the Shishugou Formation contains multiple coal streaks or thin seams, generally <1 m thick. In the upper part of the Badaowan Formation, one to two seams are present, with an average thickness of 3.95 m, although they are relatively deeply buried [64]. The organic composition of the Xishanyao Formation coals is dominated by inertinite, followed by vitrinite, with liptinite occurring in minor amounts. Vitrinite reflectance is low (0.39%–0.47%), indicating that the coals are of low rank [60,63,65].

CM rocks are distributed in the southern segment of the Huoshaoshan Anticline and along the flanks of the Xidagou Syncline axis within the Shazhang Fold-Thrust Belt. They formed through the baking of mudstones and siltstones in the roof and floor of the B seam of the Jurassic Xishanyao Formation during coal self-ignition, thereby recording high-temperature combustion metamorphism.

## 3  Sampling and analytical methods

### 3.1  Fieldwork permits

All fieldwork was carried out in compliance with local regulations. Sampling was conducted in accessible outcrops and abandoned/open mining areas, and no specific permits were required for sample collection. No protected sites or species were involved.

### 3.2  Coal maceral composition and vitrinite reflectance

Coal samples were crushed to ~1 mm, sieved, and cold-mounted in epoxy resin, then polished into coal pellets. Macerals, including vitrinite, inertinite, and liptinite and their subgroups, were identified under a reflected light microscope equipped with a 50 × oil-immersion objective. Homogeneous, crack-free vitrinite was selected for reflectance measurements, and ≥50 points were measured under monochromatic light at 546 nm. The maximum reflectance (Romax) of each point was recorded, and the mean value was calculated.

### 3.3  Proximate analysis of coal

Air-dried coal samples (<0.2 mm) were analyzed for moisture (dried at 105–110°C until constant weight), ash yield (combusted at 815 ± 10°C until constant weight), volatile matter (heated at 900 ± 10°C under an anaerobic atmosphere for 7 min, converted to dry ash-free basis, daf), and total sulfur (gravimetric sulfate precipitation). Oxygen uptake was determined using a thermogravimetric analyzer by exposing samples to dry $O_2$ at 30 °C for 48 h and recording the mass gain.

### 3.4  Ultimate analysis of coal

Air-dried coal samples (<0.2 mm) were analyzed with an elemental analyzer to determine C, H, and N by complete combustion at 1150 °C with chromatographic separation of the products. Total sulfur was measured by high-frequency

| Erathem | System | Series | Strata | Stratigraphic Column | Contact Relationship | Tectonic Movement |
|---|---|---|---|---|---|---|
| Cenozoic | Quaternary | Holocene | Loess | | | Himalayan Movement |
| | | Pleistocene | Xiyu Fm. | | Unconformity | |
| | Neogene | Miocene | Dushanzi Fm. $(N_2d)$ | | | |
| Mesozoic | Cretaceous | Lower | Tugoulu Group $(K_1T)$ | | | Yanshanian Movement (III) |
| | | | | | Unconformity | Yanshanian Movement (II) |
| | | Mid-Upper | Shishugou Group $(J_{2-3}S)$ | | | Yanshanian Movement (I) |
| | Jurassic | Middle | Xishanyao Fm. $(J_2x)$ | CM Rocks / $B_2$ / $B_1$ Coal Seam / $B_0$ $B_1$ | Unconformity | |
| | | Lower | Sangonghe Fm. $(J_1s)$ | | | |
| | | | Badaowan Fm. $(J_1b)$ | | | Indosinian Movement |
| | Triassic | Mid-Upper | Xiaogouqun Group $(T_{2-3}x)$ | | Unconformity | |

Fig 2. Stratigraphic column of the Shazhang Fold-Thrust Belt [15].

combustion at 1350 °C in an oxygen-rich atmosphere, and the released $SO_2$ was quantified using an infrared absorption detector.

### 3.5 Petrographic analysis of CM rocks

Eight representative samples of CM rocks (including clinker, paralava, and breccia) were selected for thin section preparation and petrographic analysis. Rock slices (~0.05 mm thick, ~25 × 15 mm) were cut, mounted on glass slides, and ground using progressively finer abrasives. The polished thin sections were examined and photographed under a Nikon LV100POL polarizing microscope.

### 3.6 X-ray diffraction (XRD) analysis

Ten CM rock samples with varying metamorphic degrees were ground in an agate mortar to <200 mesh and dried at 105 °C for 2 h. The powdered samples were analyzed using a Rigaku Ultima IV diffractometer equipped with Cu-Kα radiation and a D/teX Ultra detector. X-ray intensities were recorded over a 2θ range of 5–50° at a scanning rate of 20°/min under operating conditions of 40 kV and 40 mA.

### 3.7 Zircon (U-Th)/He dating

One sample of CM rock was collected, characterized by a reddish to dark-red color with vesicular textures and weak melting features, indicating a relatively high degree of combustion metamorphism. The sample was crushed, and heavy and light mineral fractions were separated using a water-shaking table, followed by heavy liquid and magnetic separation. Zircon crystals were handpicked from the mineral separates under a binocular microscope.

(U-Th)/He analyses were carried out at the University of Melbourne following the protocol of House et al. (2000) for laser He extraction from single zircon grains [66]. Clear, inclusion-free, non-fractured euhedral grains with comparable size ranges were selected, immersed in ethanol, and examined under polarized light to exclude grains with visible inclusions. All dated grains are euhedral zircons with two terminations (Table 2). Grain geometries were imaged and measured for α-ejection corrections [67], and the grains were then loaded into small acid-treated platinum capsules.

Helium was extracted by laser heating at ~12.6 W (~1300 °C) for 20 min, with hot blanks run between samples to ensure complete degassing. Degassed zircons were transferred to Parr bombs, spiked with $^{235}$U and $^{230}$Th, and digested in 0.3–0.5 mL HF at 240 °C for 40 h. Identical spiked standards and unspiked blanks were prepared for calibration. A second digestion step in HCl at 200 °C for 24 h ensured complete dissolution of fluorides. The residues were dried, dissolved in $HNO_3$, and diluted to 5% acidity with ultrapure water. U, Th, and isotopic ratios were determined using an Agilent 7700X ICP-MS.

Zircon (U-Th)/He ages were calculated and corrected for α- ejection following Farley et al [67]. Analytical accuracy was monitored using Fish Canyon Tuff zircon [68,69]. The zircon (U-Th)/He results are presented in Table 2.

## 4 Results

### 4.1 Coal characteristics

**4.1.1 Maceral composition and vitrinite reflectance.** The organic constituents of the coal are dominated by inertinite, averaging 69.52%, 70.05%, and 50.73% in the $B_2$, $B_1$, and $B_0$ seams, respectively. Vitrinite is secondary, accounting for 25.76%, 26.43%, and 37.89% in the same seams, with minor liptinite present. The maximum vitrinite reflectance (R°max) of the $B_2$ seam ranges from 0.30% to 0.55% (mean 0.40%); that of the $B_1$ seam ranges from 0.27% to 0.65% (mean 0.39%); and that of the $B_0$ seam ranges from 0.49% to 0.50% (mean 0.50%). Coals from the $B_2$, $B_1$, and $B_0$ seams of the Xishanyao Formation are classified as low-rank. Among the inorganic constituents, clay minerals are overwhelmingly predominant (Table 1).

**Table 1. Statistical results of coal petrographic identification of the B-coal seams in the Xishanyao Formation ($\frac{Min-Max}{Mean}$).**

| | Maceral composition (%) | | | | Industrial characteristics (%) | | | | | Elemental Composition (%) | | | |
|---|---|---|---|---|---|---|---|---|---|---|---|---|---|
| | Vitrinite | Inertinite | Liptinite | Vitrinite reflectance $R°_{max}$ | Mad | Ad | Vdaf | St, d | OUC (cm³/g) | Cdaf | Hdaf | Ndaf | St, d |
| B2 coal seams | 3.2-58.8 / 25.76 | 40.1-96.0 / 69.52 | 0.0-4.3 / 1.25 | 0.30-0.55 / 0.40 | 5.20-21.42 / 12.59 | 4.36-24.43 / 10.35 | 29.89-37.51 / 33.04 | 0.40-1.58 / 0.71 | 0.63-1.22 / 1.01 | 53.75-84.91 / 78.96 | 2.08-4.83 / 3.52 | 0.59-2.98 / 0.78 | 0.12-2.27 / 0.71 |
| B1 coal seams | 1.6-63.9 / 26.43 | 34.0-97.4 / 70.08 | 0-4.93 / 1.06 | 0.27-0.65 / 0.39 | 5.14-19.92 / 10.75 | 5.24-22.39 / 9.09 | 29.43-38.26 / 32.61 | 0.39-0.86 / 0.59 | 0.86-1.18 / 1.05 | 65.62-83.83 / 79.48 | 0.00-5.27 / 3.59 | 0.46-4.61 / 0.75 | 0.00-2.05 / 0.60 |
| B0 coal seams | 20.78-55 / 37.89 | 43.8-57.65 / 50.73 | 0.2-4.31 / 2.23 | 0.49-0.50 / 0.50 | 5.64-14.51 / 9.64 | 9.63-30.69 / 18.92 | 32.34-42.02 / 37.70 | 0.37-0.83 / 0.64 | 0.82-1.05 / 0.93 | 55.10-81.12 / 76.51 | 2.21-5.01 / 3.76 | 0.56-1.41 / 0.84 | 0.11-2.64 / 0.67 |

**4.1.2 Industrial characteristics of coal.** On an air-dried basis, moisture contents (Mad) in the $B_2$, $B_1$, and $B_0$ seams average 12.59%, 10.75%, and 9.64%, respectively, indicating ultra-low to medium moisture overall. Dry-basis ash yields (Ad) average 10.35%, 9.09%, and 18.92%, spanning ultra-low-ash to high-ash, with low-ash coals predominating. Dry, ash-free volatile matter (Vdaf) averages 33.04%, 32.61%, and 37.70%, corresponding to medium- to high-volatile coals. Dry-basis total sulfur (St,d) averages 0.71%, 0.59%, and 0.64% for $B_2$, $B_1$, and $B_0$, respectively. The mean oxygen uptake capacity (OUC) is 1.01, 1.05, and 0.93 cm³/g for $B_2$, $B_1$, and $B_0$, suggesting an elevated tendency toward spontaneous combustion.

**4.1.3 Elemental composition of coal.** Organic matter represents the principal chemical component of coal, dominated by carbon, hydrogen, nitrogen, and sulfur, with elemental abundances summarized in Table 1. On a dry, ash-free basis, the mean carbon contents (Cdaf) of the $B_2$, $B_1$, and $B_0$ seams are 78.96%, 79.48%, and 76.51%, respectively. The corresponding mean hydrogen contents (Hdaf) are 3.52%, 3.59%, and 3.76%, while mean nitrogen contents (Ndaf) are 0.78%, 0.75%, and 0.84%, respectively.

## 4.2 Distribution and petro-mineralogical characteristics of CM rocks

CM rocks are mainly distributed within the Shazhang Fold-Thrust Belt, occurring in the southern segment of the Huoshaoshan Anticline and on both flanks of the axis of the Xidagou Syncline, with a general SW–NE strike. The combustion metamorphic zone extends over approximately 10 km², with the depth of combustion increasing progressively from the northwest and west toward the southeast and east. The regional topography gently slopes from north to south, forming a landscape of hills and depressions (Fig 3 a-c), with elevations ranging from 487 to 538 m and a north–south relief of 51 m.

CM rocks are high-temperature metamorphic products formed by coal seam combustion. During spontaneous combustion, the adjacent sandstones or mudstones were subjected to the high-temperature baking of the burning coal seams, which altered their original characteristics and led to dehydration or oxidation, thereby modifying their petrographic, mineralogical, and geochemical properties. Based on the degree of combustion metamorphism, CM rocks can be divided into three types: baked rocks, clinker, and paralava. The most distinctive petrographic feature of CM rocks is their characteristic coloration. Clinker is typically brick-red, whereas some may also exhibit light yellow or gray-green hues (Fig 3 d-f). Paralava usually exhibits brown or grayish-black colors (Fig 3f). Due to the high-temperature roasting during coal seam combustion, CM rocks show distinct structures and textures, mainly including relic structures, vesicular structures, columnar jointing (Fig 3g), and combustion-induced fractures (Fig 3h).

Petrographic observations show that in red baked rocks, quartz grains develop numerous microcracks due to the intense thermal stress generated by rapid heating and cooling; hematite adhering to clay minerals is predominantly red (Fig 4a). The thermal stress associated with combustion metamorphism significantly enhances the undulatory extinction of quartz, making it more widespread, pronounced, and complex, with broader and more irregular extinction bands (Fig 4b). In ochre-red clinkers, the edges of quartz grains are partially melted, becoming rounded and smooth, and in some cases

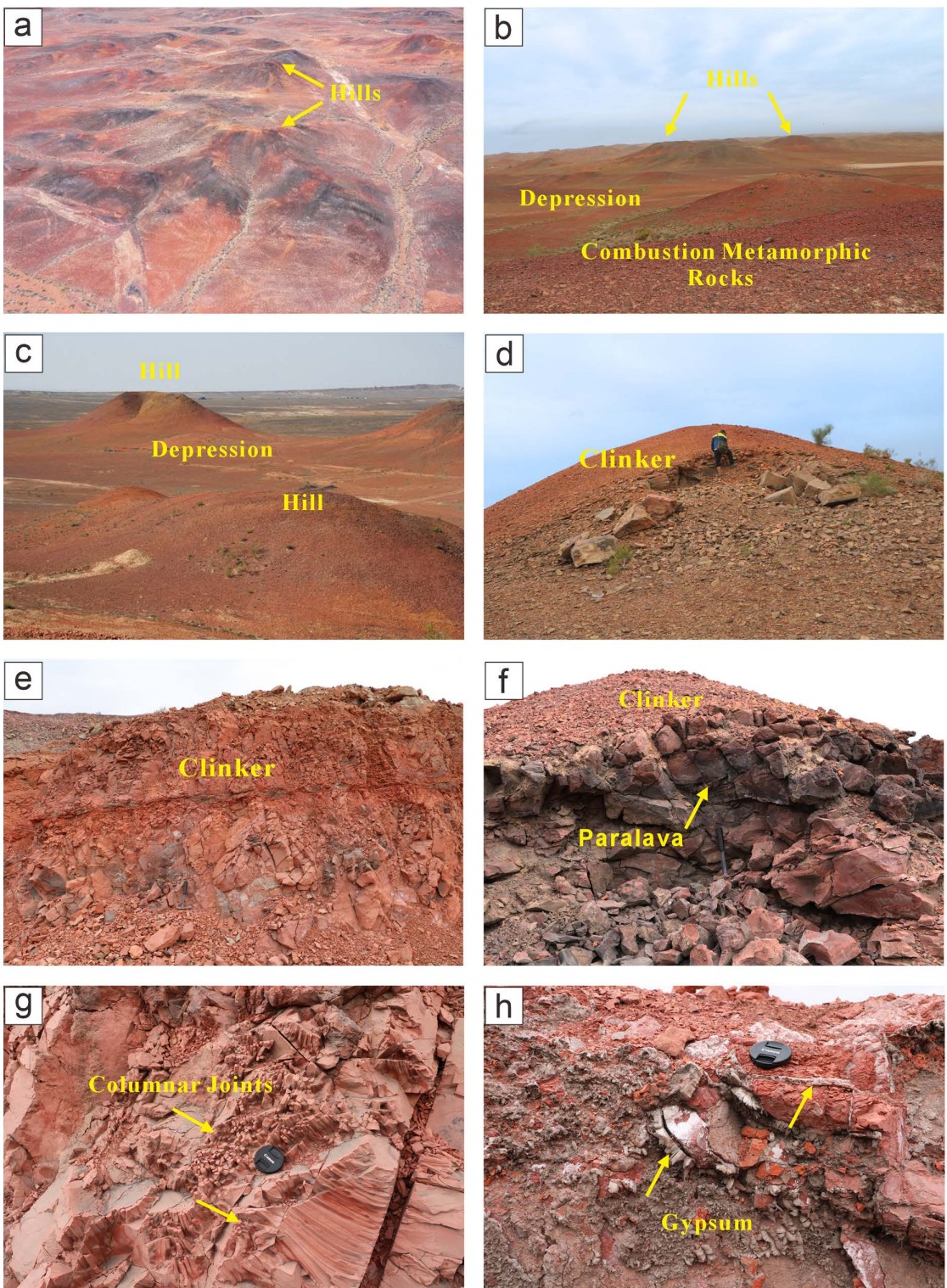

**Fig 3. Field photographs of CM rocks. (a-c)** Hills and depressions formed by CM rocks. **(d)** Hill formed by CM rocks, more than 10 m high. **(e)** Brick-red clinker. **(f)** Upper part composed of brick-red clinker and lower part of grayish-black paralava. **(g)** Columnar joints developed in CM rocks. **(h)** Gypsum infilling within fractures of CM rocks.

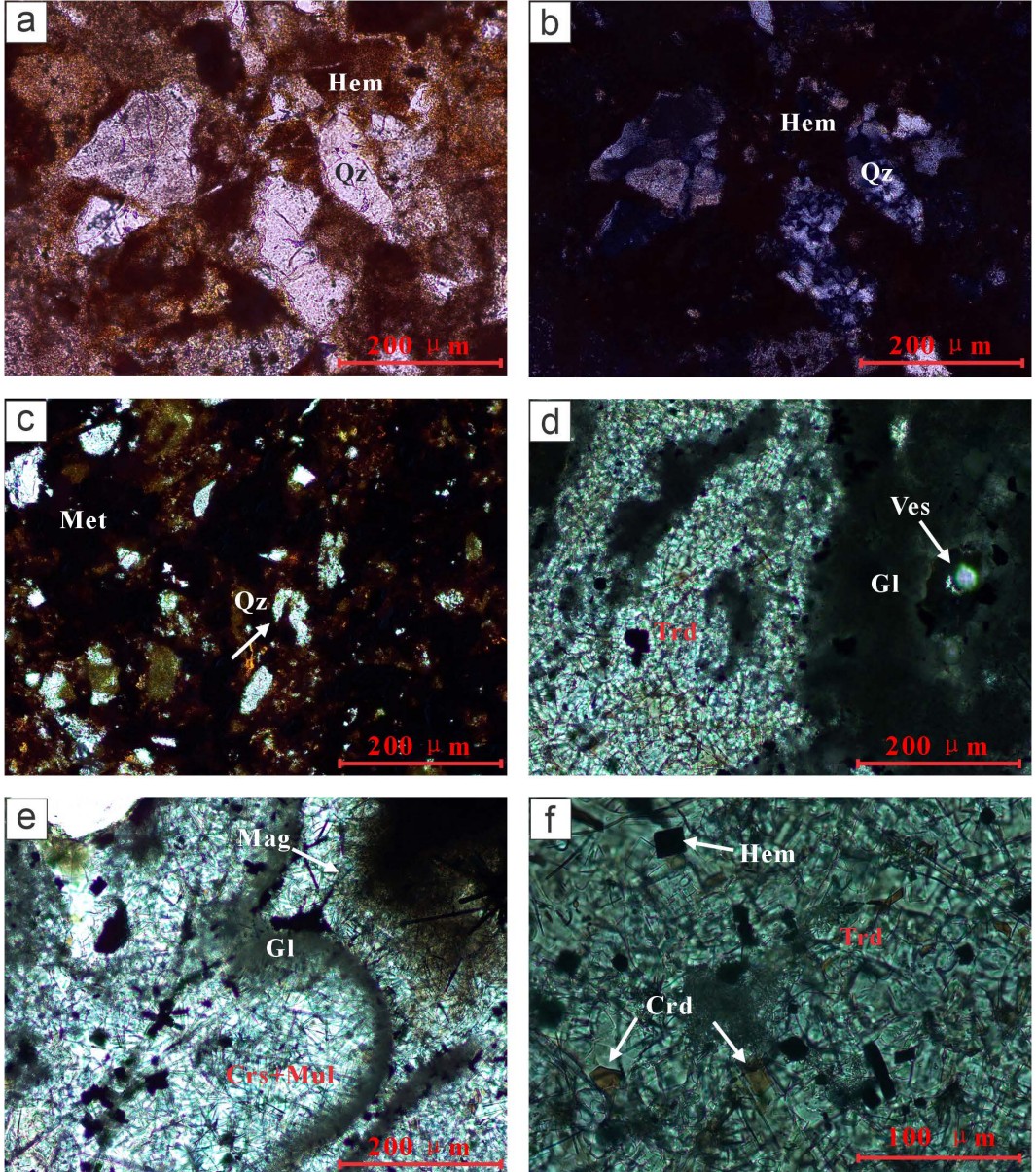

**Fig 4. Thin-section micrographs of CM rock.** **(a)** Red clinker, quartz grains with microcracks, clay minerals coated by hematite (PPL). **(b)** Red clinker, quartz grains showing undulatory extinction (XPL). **(c)** Reddish-brown clinker, quartz grains with embayed margins, iron oxides coating or disseminated within the glassy matrix, appearing opaque (PPL). **(d)** Dark-gray paralava, tridymite, glass, vesicles (PPL). **(e)** Paralava, dendritic magnetite, radiating/ fibrous intergrowths of cristobalite and mullite (PPL). **(f)** Paralava, rhombohedral hematite, tridymite, cordierite (PPL). Quartz, Qz; Hematite, Hem; Tridymite, Trd; Cordierite, Crd; Mullite, Mul; Glass, Gl; Vesicles, Ves.

exhibiting embayed contours; iron oxides appear opaque, with transparent edges showing dark brown coloration (Fig 4c). In dark gray paralava, tridymite is observed, and the glassy matrix appears black due to the presence of iron minerals, with vesicles present within the glass (Fig 4d). Additionally, mullite and cristobalite form radially oriented symplectitic aggregates, and dendritic magnetite and cordierite are visible (Fig 4 e, f).

XRD results show that in red clinker, the dominant minerals are quartz, hematite, and ferrocordierite (Fig 5a). In paralava, the main minerals include quartz, microcline, albite, phosphoquartz, and pseudobrookite (Fig 5b). Some of these minerals are products of high-temperature metamorphic reactions during coal seam combustion.

### 4.3 (U-Th)/He ages

Mineral separation was conducted on red clinker sample, and five zircon grains were selected for (U-Th)/He dating. Three zircons yielded similar ages, ranging from 0.56±0.03 to 0.70±0.04 Ma, with a weighted mean age of 0.63±0.19 Ma. Two additional grains yielded markedly older ages of 8.7±0.5 Ma and 87.7±5.4 Ma (Table 2). All obtained ages are younger

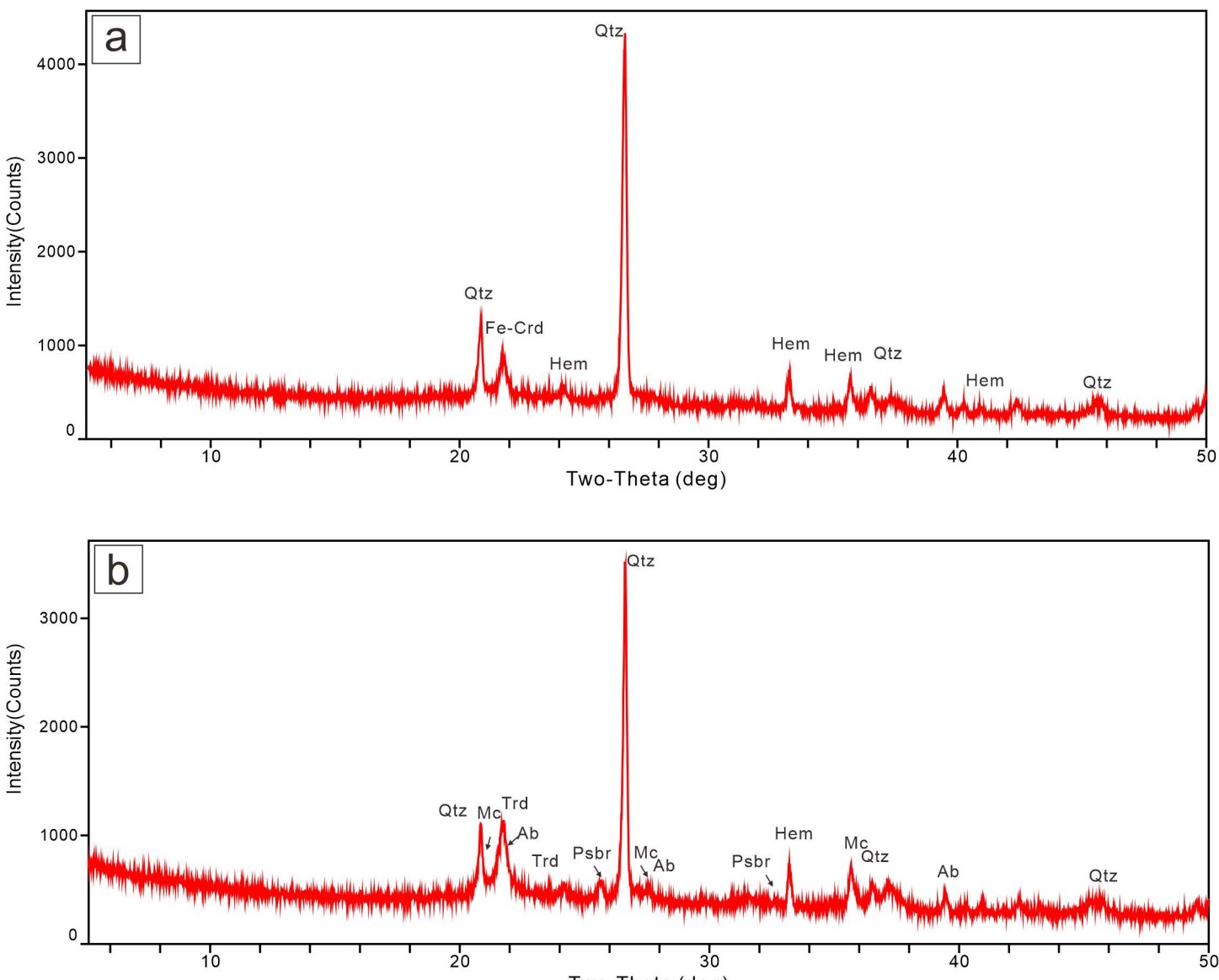

**Fig 5. XRD diagrams of CM rocks.** (a) clinker, **(b)** Paralava. Quartz, Qtz; Tridymite, Trd; Hematite, Hem; Ferrocordierite, Fe-Crd; Microcline, Mc; Albite, Ab; Pseudobrookite, Psbr.

**Table 2. Single grain zircon (U-Th)/He ages of CM rocks.**

| Sample | He# | ⁴He (ncc) | Mass (mg) | ᵃFₜ | U ppm | Th ppm | Th/U ratio | ᵇ[eU] ppm | Corrected age (Ma) | Error (±1) | Grain length (m) | Grain half width (m) | ᶜRs | ᵈCrystal morphology |
|--------|-----|-----------|-----------|-----|-------|--------|------------|-----------|---------------------|-----------|------------------|----------------------|-----|---------------------|
| HSS-1 | 69016 | 0.035 | 0.0012 | 0.64 | 293.6 | 237.8 | 0.81 | 349.4 | 0.70 | 0.04 | 109.9 | 31.9 | 37.0 | 2T |
| HSS-1 | 69022 | 0.049 | 0.0018 | 0.66 | 296.6 | 181.6 | 0.61 | 339.3 | 0.67 | 0.04 | 160.8 | 28.3 | 36.1 | 2T |
| HSS-1 | 69028 | 0.030 | 0.0030 | 0.69 | 118.0 | 117.1 | 0.99 | 145.5 | 0.56 | 0.03 | 206.0 | 31.8 | 41.3 | 2T |
| Weighted mean 0.63 ± 0.19 | | | | | | | | | | | | | | |
| HSS-1 | 70312 | 2.762 | 0.0041 | 0.75 | 565.6 | 277.5 | 0.49 | 630.8 | 8.7 | 0.5 | 170.3 | 46.6 | 54.9 | 2T |
| HSS-1 | 69019 | 9.782 | 0.0032 | 0.73 | 269.1 | 67.1 | 0.25 | 284.9 | 87.7 | 5.4 | 185.6 | 35.9 | 45.1 | 2T |

ᵃ FT is the a-ejection correction after Farley et al. (1996).

ᵇ Effective uranium concentration (U ppm + 0.235 Th ppm).

ᶜ Equivalent spherical radius ([Rs] = [3*R*L]/[2*[R + L]] after Beucher et al. (2013).

ᵈ Grain morphology - 0T = no terminations, 1T = one termination, 2T = 2 terminations.

than the depositional age of the host strata, suggesting that the samples experienced post-depositional thermal annealing. An eU-age comparison shows no simple systematic negative correlation between age and eU, suggesting that radiation damage alone is insufficient to explain the observed dispersion.

## 5 Discussion

### 5.1 Causes of coal seam spontaneous combustion

Coal seam spontaneous combustion is controlled by multiple factors, which can be broadly categorized into three aspects: the intrinsic propensity of the coal to spontaneous combustion, the thickness and occurrence of the coal seam, and external environmental conditions [70–73].

The intrinsic propensity of coal to spontaneous combustion is considered an internal factor, mainly governed by its rank, maceral composition, moisture, ash yield, volatile matter, total sulfur, and oxygen absorption capacity [74–77]. The tendency of coal to self-ignite is generally inversely proportional to its rank, with low-rank coals being more susceptible to spontaneous combustion than high-rank coals [78–80]. Inertinite, characterized by a loose structure and strong oxygen affinity, enhances the susceptibility of coal seams to self-heating [81]. In addition, high volatile matter, high moisture, elevated contents of carbon, hydrogen, and nitrogen, and reduced ash yield collectively contribute to an increased tendency for spontaneous combustion [73,78]. Smaller coal particle sizes further promote self-heating, as their larger specific surface area enhances both oxygen accessibility and heat transfer, thereby increasing the risk of self-ignition [82–84].

In the study area, the spontaneous combustion tendency of coal seams was investigated. The Jurassic Xishanyao Formation B-seam coals are dominated by inertinite, followed by vitrinite and liptinite. The vitrinite reflectance (Rᵒmax, %) ranges from 0.39 to 0.50 (Table 1), indicating a low-rank coal with a high propensity for spontaneous combustion. In addition, the B-seam coals of the Xishanyao Formation are characterized by high moisture, high volatile matter, large specific surface area, and low ash yield. The elemental compositions of carbon, hydrogen, nitrogen, and sulfur also show a strong linear relationship with the spontaneous combustion tendency index (Table 1). Therefore, the B-seam coals of the Xishanyao Formation exhibit a pronounced tendency toward spontaneous combustion. The thickness of a coal seam provides the material basis for spontaneous combustion. Thicker seams generate more heat during oxidation, and the heat is less likely to dissipate, thereby increasing the likelihood of spontaneous combustion [50,72]. In the study area, the coal seams of Member 3 of the Xishanyao Formation have an average thickness of 2–26 m, corresponding to medium to thick seams, which favors spontaneous combustion.

In addition to intrinsic factors such as the spontaneous combustion tendency, thickness, and occurrence of coal seams, external conditions mainly include ignition sources (e.g., lightning strikes or wildfires), arid climatic conditions, and the

availability of oxygen [35,85]. Previous studies have further shown that coal seam spontaneous combustion occurs more frequently under arid to semi-arid climatic conditions, where low precipitation, sparse vegetation, and windy environments favour long-term exposure of coal seams and sustained oxidation and heat accumulation. In contrast, during more humid climatic stages, higher groundwater levels and thicker overburden tend to suppress coal seam spontaneous combustion to some extent [35,36,78,86].

Nevertheless, the most critical external factor controlling the initiation and persistence of coal seam spontaneous combustion is the availability of sufficient oxygen, which constitutes the prerequisite for coal self-ignition [87,88]. Regional tectonic uplift and denudation can raise deeply buried coal seams to shallow levels or even expose them at the surface, thereby providing the oxygen required for combustion. Consequently, such geological processes play a decisive role in triggering coal seam spontaneous combustion [38,44,89]. In the study area, tectonic uplift and erosion since the Late Cenozoic have progressively brought coal seams to shallow depths or surface exposure. Meanwhile, faults and joints generated by tectonic activity cut through the coal seams, enhancing downward oxygen migration from the surface into the coal-bearing strata, promoting oxidation reactions and ultimately initiating spontaneous combustion.

## 5.2  Coupling between tectonic uplift events and the formation of CM rocks

Tectonic uplift and denudation are the key external processes that bring coal seams into shallow, oxygen-rich conditions and thereby enable spontaneous combustion. Since the Mesozoic, the eastern orogenic belt of the Junggar Basin has undergone multiple phases of uplift and erosion [19,22,90]. Within this framework, the zircon (U-Th)/He ages of CM rocks provide direct chronological information on the timing at which the Xishanyao coal seams approached or reached near-surface conditions in the study area.

The dominant zircon (U-Th)/He ages of CM rock samples obtained in this study range from $0.56 \pm 0.03$ to $0.70 \pm 0.04$ Ma, with a weighted mean age of $0.63 \pm 0.19$ Ma, is interpreted as the principal age of coal-seam combustion and CM rock formation in the Middle Pleistocene (Table 2). This result indicates that the Jurassic Xishanyao Formation B coal seams were uplifted and eroded to near-surface levels during the Middle Pleistocene, subsequently undergoing spontaneous combustion and forming CM rocks, during which the zircon (U-Th)/He system in most grains was fully reset. In contrast, the two older single-grain ages ($8.7 \pm 0.5$ Ma and $87.7 \pm 5.4$ Ma) do not define coherent additional age populations. Given their single-grain occurrence, large deviation from the clustered population and the absence of a systematic eU-age trend, these two grains are more conservatively interpreted as incompletely reset or inherited pre-combustion thermochronologic components rather than as evidence for separate combustion episodes.

This interpretation should nevertheless be regarded as cautious rather than definitive. The dated grains are all euhedral and free of optically visible inclusions under binocular and polarized-light inspection, but no cathodoluminescence imaging or internal U-Th zoning analyses were carried out. Accordingly, crystal-scale heterogeneity cannot be fully excluded. Likewise, because the dataset is limited to five usable single-grain ages from one sample, it is not sufficiently robust for inverse thermal-history modelling, and the distinction between partial resetting and inheritance cannot be resolved uniquely. Future studies based on larger single-grain datasets, together with zircon imaging and complementary thermochronometers, will be important for refining the interpretation of the anomalously old grains.

The spatial distribution and geological context of the CM rocks nonetheless support a close coupling between uplift and combustion. CM rocks occur in the central part of the Shazhang Fold-Thrust Belt, specifically on the southern segment of the Huoshaoshan Anticline and along both flanks of the Xidagou Syncline axis. The overall stratigraphic thickness of the Xishanyao Formation shows a decreasing trend near the Xidagou Syncline and the axis of the Huoshaoshan Anticline, indicating that the anticline had already developed prior to coal seam deposition and subsequently experienced uplift. During the Late Permian, nearly east–west compressive stress modified the region, giving rise to a series of incipient north–south-trending folds. In the Middle Jurassic, during the deposition of the Xishanyao Formation, the intense

Yanshanian orogeny drove north–south plate convergence, resulting in uneven uplift and subsidence in the Shazhang area [91,92]. In the Late Jurassic, nearly east–west shortening further intensified the north–south folds formed in the Late Permian, eventually establishing their final structural pattern and producing an angular unconformity between the Cretaceous and Jurassic strata [12,92]. Subsequently, the study area underwent multiple episodes of uplift and denudation, including during the Late Cretaceous–Paleocene and since the Miocene [20]. These earlier structural developments set the stage for later exhumation of the coal-bearing strata.

Critically, the rapid uplift and denudation phases since the Miocene are temporally linked with the age signals identified in the CM rocks. During this period, the Bogda Mountain and Kalamaili Mountain experienced intense uplift due to north–south compression triggered by the far-field effects of the collision between the Indian and Eurasian plates. Liu (2023) suggested that this stage was characterized by the greatest magnitude of uplift, the most intense tectonic activity, and the strongest deformation of strata along the basin margins [19]. Apatite fission-track dating of volcanic rocks near Tianchi on the northern flank of the Bogda Mountains indicated an uplift–denudation phase during 12–7 Ma, as revealed by quantitative thermal history modeling [9]. Similarly, apatite fission-track dating of tuffaceous sandstones from the northern Bogda region demonstrated a rapid uplift–denudation phase since 11 Ma, with an estimated uplift rate of 190.6 m/Ma [23]. In the Kelameili Mountains, studies have shown that the most pronounced phase of rapid uplift–denudation occurred since ~20 Ma, with an uplift rate of approximately 71 m/Ma [22,93]. Thermal history modeling of samples from the basin margins further indicates that nearly all of them recorded a rapid uplift phase since the Late Miocene (ca. 10 Ma) (Fig 6).

In this regional context, the ~8.7 Ma zircon (U-Th)/He age is compatible with retention of an older uplift-related cooling signal in a grain that was not completely degassed during the later combustion event. Such partial resetting and preservation of inherited zircon (U-Th)/He ages are common in transient high-temperature systems, including CM rocks [38,45]. The much older 87.7 Ma grain is best regarded as a residual pre-combustion age component that has no direct significance for the timing of coal burning at the study site.

Continued tectonic activity and erosion after the Late Miocene eventually culminated in the Middle Pleistocene exposure of the Jurassic coal seams. The clustered ~0.63 Ma ages therefore record the stage at which the coal seams actually reached near-surface, oxygenated conditions and spontaneous combustion was initiated. In this sense, the Late Miocene to Pliocene uplift prepared the structural and geomorphic framework for combustion, whereas Middle Pleistocene exhumation provided the final condition necessary for ignition. The present study thus suggests a sequential relationship between regional tectonic exhumation and subsequent CM rock formation, with a measurable lag between broader orogenic uplift and local combustion metamorphism.

In addition, the main faults in the study area include the Shaxi Fault, Huodong Fault, and Zhangdong Fault. All of them belong to basement-involved triangle shear faults, which initiated in the late Permian and were finally formed at the end of the Cretaceous. The fault terminations only extend into the Lower Jurassic strata and do not cut through the Jurassic coal seams [11]. Therefore, we infer that these faults did not provide favorable conditions for coal seam spontaneous combustion. The combustion ages therefore appear to represent a lagged response to regional uplift and erosion rather than direct fault-controlled ignition. Such a lag is reasonable because tectonic stress was first accommodated by uplift of the surrounding mountain belts and only later translated into sufficient local exhumation to expose coal seams at the surface or near surface.

In summary, the formation ages of the CM rocks record a sequential coupling between regional tectonics exhumation and local coal-seam combustion. The dominant Middle Pleistocene age of ~0.63 Ma tightly constrains the timing at which continued uplift and erosion exposed the Jurassic coal seams to near-surface, oxygen-rich conditions, thereby triggering spontaneous combustion and CM rock formation. The isolated older ages are interpreted more cautiously as incompletely reset or inherited components that may retain information on earlier exhumation history, but they are not used as direct evidence for separate coal-fire events in the study area.

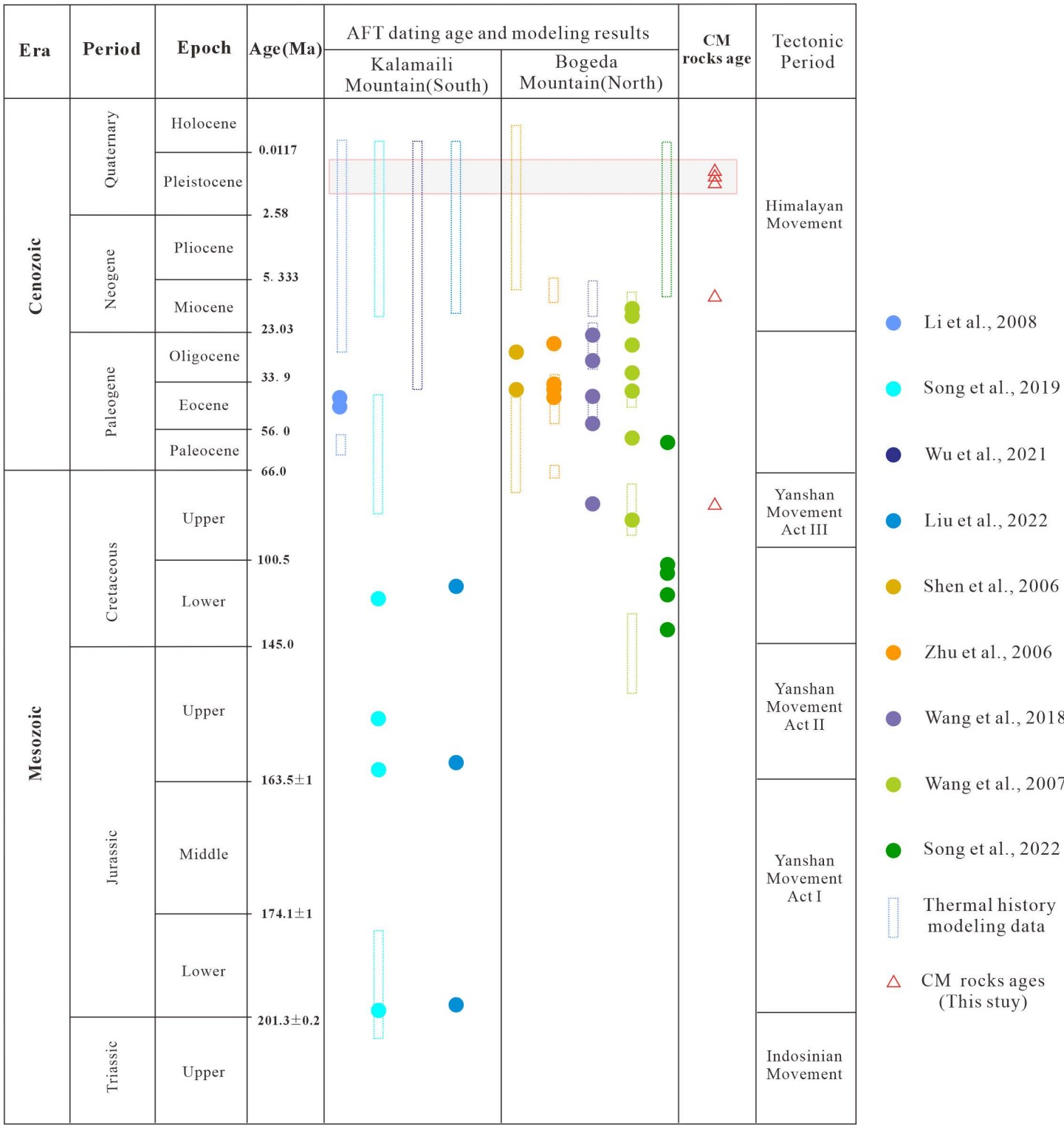

**Fig 6. Statistical summary of Mesozoic-Cenozoic tectonic activity timings [2,3,9,14,17,22,23,93,94] and CM rock ages in the orogenic belts around the eastern Junggar Basin.**

## 6 Conclusions

(1) The Jurassic Xishanyao Formation coals are dominated by inertinite, exhibit low vitrinite reflectance, and contain high volatile matter and moisture, indicating a strong tendency toward spontaneous combustion. Following uplift and exposure, these coals generated CM rocks – baked rocks, clinker, and paralava – characterized by distinctive petrographic features (e.g., vesicular textures, columnar joints) and high-temperature mineral assemblages (e.g., hematite, tridymite, and mullite).

(2) Zircon (U-Th)/He dating of CM rocks identifies a dominant Middle Pleistocene age with a weighted mean of ~0.63 Ma. this age cluster is interpreted as the principal timing of coal seam uplift and exhumation to near-surface exposure of the Xishanyao coal seams. Two older isolated ages are treated cautiously as incompletely reset or inherited components rather than as independent combustion events.

(3) The spatiotemporal distribution of CM rocks is closely linked to late Cenozoic uplift and denudation of the eastern Junggar orogenic belt. Accordingly, zircon (U-Th)/He thermochronology of CM rocks provides a useful supplementary approach for constraining the timing of near-surface tectonic exhumation in regions where conventional chronometers have limited resolution.

## Supporting information

**S1 Table. Statistical results of coal petrographic identification of the B-coal seams in the Xishanyao Formation.** (XLSX)

## Acknowledgments

We would like to thank all those who supported us during our fieldwork in the Junggar region. Our special thanks go to Barry P. Kohn from the School of Earth Sciences, University of Melbourne, for his valuable assistance with the zircon (U-Th)/He analyses. We are also deeply grateful to Panjie Zhang from Yankuang Xinjiang Energy & Chemical Co., for his support with the coal analyses. We appreciate the constructive comments and suggestions provided by the anonymous reviewers, which greatly improved the quality of this manuscript.

## Author contributions

**Conceptualization:** Bin Chen.

**Data curation:** Pan Liu, Zhuang Zhao.

**Formal analysis:** Pan Liu.

**Funding acquisition:** Bin Chen.

**Investigation:** Pan Liu, Yan Dong, Changjuan Feng, Zhuang Zhao.

**Methodology:** Bin Chen.

**Project administration:** Bin Chen.

**Software:** Yan Dong.

**Validation:** Bin Chen, Yan Dong, Chaoqun Yang, Yixin Dong.

**Writing – original draft:** Bin Chen.

**Writing – review & editing:** Bin Chen.

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
