## [Decision Letter · Decision Letter 0]

18 Nov 2025

PONE-D-25-53522Late cenozoic exhumation in the eastern Junggar Basin: Evidence from zircon (U-Th)/He ages of combustion metamorphic rocksPLOS ONE

Dear Dr. Chen,

Thank you for submitting your manuscript to PLOS ONE. After careful consideration, we feel that it has merit but does not fully meet PLOS ONE’s publication criteria as it currently stands. Therefore, we invite you to submit a revised version of the manuscript that addresses the points raised during the review process.

We look forward to receiving your revised manuscript.

Kind regards,

Hu Li

Academic Editor

PLOS ONE

Journal Requirements:

“This research was funded by the National Natural Science Foundation of China, under grant number 42201006; by the Open Fund (DGERA 20221105) of the Key Laboratory of Deep-time Geography and Environment Reconstruction and Applications of the Ministry of Natural Resources, Chengdu University of Technology;”

4. We note that your Data Availability Statement is currently as follows: [All relevant data are within the manuscript.]

“This research was financially supported by the National Natural Science Foundation of China (Grant No. 42201006). Additional support was provided by Open Fund (DGERA 20221105) of Key Laboratory of Deep-time Geography and Environment Reconstruction and Applications of Ministry of Natural Resources, Chengdu University of Technology.”

“This research was funded by the National Natural Science Foundation of China, under grant number 42201006; by the Open Fund (DGERA 20221105) of the Key Laboratory of Deep-time Geography and Environment Reconstruction and Applications of the Ministry of Natural Resources, Chengdu University of Technology;”

Reviewers' comments:

Reviewer's Responses to Questions

**Comments to the Author**

1. Is the manuscript technically sound, and do the data support the conclusions?

Reviewer #1: Partly

Reviewer #2: Yes

2. Has the statistical analysis been performed appropriately and rigorously? 

Reviewer #1: No

Reviewer #2: N/A

3. Have the authors made all data underlying the findings in their manuscript fully available?

Reviewer #1: Yes

Reviewer #2: Yes

4. Is the manuscript presented in an intelligible fashion and written in standard English?

Reviewer #1: Yes

Reviewer #2: Yes

5. Review Comments to the Author

Reviewer #1: Reviewer Comments on “Late Cenozoic exhumation in the eastern Junggar Basin: Evidence from zircon (U-Th)/He ages of combustion metamorphic rocks”

This manuscript investigates the Late Cenozoic exhumation of the eastern Junggar Basin by applying zircon (U-Th)/He dating to combustion metamorphic (CM) rocks formed through spontaneous combustion of coal seams during uplift and denudation. The authors propose that the formation ages of CM rocks provide direct temporal constraints on the timing of near-surface exposure and thus the tectonic evolution of the adjacent orogenic belt. This is an interesting approach that may open new perspectives for near-surface thermochronology in tectonically active regions. In general, the current version provides an intriguing idea but lacks the depth of quantitative and geological validation required for publication in its present form. Below I list my major concerns:

1. The paper assumes that the combustion process completely resets the zircon (U-Th)/He system. However, CM rocks typically experience very high temperatures (up to >1000 °C) but over short durations (hours to days), leading to highly non-equilibrium diffusion conditions. The authors should provide a more quantitative discussion on whether such transient heating events can fully reset zircon He ages. Incorporating diffusion models or referencing kinetic studies would help justify the reliability of the obtained ages.

2. The manuscript reports a Middle Pleistocene (~0.63 Ma) age and additional Late Miocene ages. It is unclear whether these reflect multiple combustion episodes, partial resetting, or sample mixing. A more detailed explanation is needed on how these age populations were identified, and what geological processes they represent. In particular, how can the Late Miocene event be reconciled with the much younger Pleistocene age - do they correspond to different tectonic stages, or to local reactivation and erosion events?

3. While CM rock formation is indeed associated with near-surface exposure, it may also depend on factors such as coal rank, oxygen supply, aridity, and fracture permeability. Therefore, linking combustion ages directly to tectonic uplift could oversimplify the process. The authors should discuss possible non-tectonic controls and clarify how they distinguished tectonic from environmental triggers.

4. The regional tectonic evolution of the eastern Junggar belt should be described more comprehensively. How do the derived ages compare with other thermochronological or geomorphic constraints from the nearby Bogda Shan, Kelameili, or Karamay regions? Including a regional synthesis (e.g., with published AFT, AHe, or sedimentary record data) would enhance the geological significance of this study.

Reviewer #2: Abstract: Please add more details regarding the data and results.

Line 53: The abbreviation is not necessary.

Line 91: please add one or two sentence to introduce the tectonic setting or the regional geology of the Junggar basin, such as: https://doi.org/10.1007/s12371-019-00346-5;
https://doi.org/10.1080/00387010.2020.1792504;
https://doi.org/10.1007/s12371-024-01056-3

Line 101: “it”, please specify;

Line 142: Which study area? Please specify clearly.

Line 198: Section 4.1.1

Line 206: Section 4.1.2

Line 215: Section 4.1.3

Section 4.3: “and another 87.7 ± 5.4 Ma; due” — how should this be interpreted? Why was this data point discarded?

Section 5.1: Are these strata currently undergoing spontaneous combustion, or did combustion occur in the Cenozoic? Low-temperature thermochronologic ages are affected by closure temperature. The basic assumption is that during uplift and exhumation, as sample depth decreases or the geothermal gradient decreases, the temperature drops below the closure temperature and the thermochronologic clock starts recording. If the samples in this study experienced combustion, would that affect the closure temperature? In particular, if the combustion temperature exceeded the closure temperature, would that imply that the thermochronologic ages might not reflect uplift or exhumation?

Section 5.1, second-to-last paragraph: This paragraph might be merged with the preceding or following paragraph. With only two sentences, it does not read as a complete paragraph.

Figure 1: Axial traces are generally shown in black, and faults in red. Please also add representative attitudes (strike and dip) of strata.

Figure 2: Which study area? Please specify.

Figure 6: More detailed and case-specific interpretation is needed. For example, low-temperature thermochronology ages do not necessarily directly represent the timing of tectonic events.

Figure 7: If these data were not collected in this study and are not presented as part of the study’s dataset, they should be moved to the Geological Background section.

Section 6: A subheading (1) is not necessary. This sentence can serve directly as the topic sentence of the following paragraph.

6. PLOS authors have the option to publish the peer review history of their article (what does this mean?). If published, this will include your full peer review and any attached files.

Reviewer #1: **Yes:**Zhiyuan He

Reviewer #2: **Yes:**Liang Qiu

---

## [Author Response · Author response to Decision Letter 1]

25 Dec 2025

Response to academic editor

We thank you for the editorial evaluation and for the opportunity to revise our manuscript. We have carefully addressed all additional journal requirements requested by PLOS ONE. Our point-by-point responses are provided below.

Requirement 1

1 Please ensure that your manuscript meets PLOS ONE's style requirements, including those for file naming.

Response:

We have revised the manuscript to comply with PLOS ONE formatting and style requirements (including section structure, headings, and overall presentation). We will also follow PLOS ONE’s file naming requirements upon resubmission and upload the revision files as instructed (Response to Reviewers; Revised Manuscript with Track Changes; Clean Manuscript).

Requirement 2

In your Methods section, please provide additional information regarding the permits you obtained for the work. Please ensure you have included the full name of the authority that approved the field site access and, if no permits were required, a brief statement explaining why.

Response:

We have added a dedicated subsection in the Methods titled “Fieldwork permits”. In this subsection we clarify that no specific permits were required for field access and sampling, because the work was conducted at accessible outcrops/abandoned or open areas without involving protected sites, protected species, or restricted localities, and all activities complied with local regulations.

(Section 3.1 “Fieldwork permits.”)

Requirement 3

Response:

We confirm that the funding agencies had no role in the study design, data collection and analysis, decision to publish, or preparation of the manuscript.

Requirement 4

Data Availability Statement: please confirm whether submission contains all raw data required to replicate results (“minimal dataset”). If not, upload as Supporting Information or deposit to a repository.

Response:

We confirm that the submission will include the minimal dataset required to replicate all findings in the study. The key underlying numerical data used in our analyses and interpretations (including coal analytical results and single-grain zircon (U–Th)/He data) are provided in the manuscript tables. In addition, to fully comply with PLOS ONE’s minimal dataset policy, we have deposited the Supporting Information file Table S1. Statistical results of coal petrographic identification of the B-coal seams in the Xishanyao Formation in the Figshare repository.

Requirement 5

Please remove any funding-related text from Acknowledgments and tell us how you would like to update your Funding Statement.

Response:

We have removed all funding-related statements from the Acknowledgments section in the revised manuscript. Funding information will be provided only in the Funding Statement in the online submission system.

Funding Statement (requested update / confirmation):

We would like the Funding Statement to read as follows:

“This research was funded by the National Natural Science Foundation of China (Grant No. 42201006) and by the Open Fund (DGERA 20221105) of the Key Laboratory of Deep-time Geography and Environment Reconstruction and Applications of the Ministry of Natural Resources, Chengdu University of Technology. We confirm that the funding agencies had no role in the study design, data collection and analysis, decision to publish, or preparation of the manuscript.” Please help us submit it via the online.

Requirement 6

Response:

We have evaluated all reviewer-suggested references and incorporated those that are relevant to strengthening the manuscript’s regional tectonic and geological background and interpretation. References not directly relevant were not added.

Response to Reviewer #1

We thank you for the constructive and insightful comments. We have revised the manuscript accordingly. Major revisions include: (i) expanding the discussion of transient high-temperature heating and its implications for zircon (U–Th)/He resetting; (ii) clarifying the interpretation of the Middle Pleistocene age cluster versus isolated older ages and explaining how these age populations are identified and interpreted; (iii) adding discussion of non-tectonic controls on coal-seam combustion (coal rank, oxygen supply, aridity, ignition sources) and clarifying how tectonic vs. environmental triggers are treated; and (iv) strengthening the regional tectonic synthesis by comparing our ages to published thermochronologic constraints from nearby Bogda Shan.

Comment 1

The paper assumes that the combustion process completely resets the zircon (U-Th)/He system. However, CM rocks typically experience very high temperatures (up to >1000 °C) but over short durations (hours to days), leading to highly non-equilibrium diffusion conditions. The authors should provide a more quantitative discussion on whether such transient heating events can fully reset zircon He ages. Incorporating diffusion models or referencing kinetic studies would help justify the reliability of the obtained ages.

Response:

We agree that transient heating and non-equilibrium conditions are important when interpreting zircon (U–Th)/He ages from combustion metamorphic (CM) settings. In the revised manuscript, we expanded the discussion of reset/partial-reset behavior under short-duration, high-temperature heating by incorporating empirical/kinetic constraints and published applications of ZHe in CM rocks.

Specifically, we added text noting (i) combustion temperatures can exceed 1000 °C and may persist for long durations in natural coal-seam fires (reported to last up to hundreds of years), which increases the likelihood of substantial He loss; and (ii) empirical studies of brief, high-temperature heating (e.g., wildfire analogs) demonstrate strong He loss in apatite (>90%) but more limited He loss in zircon (~15%), highlighting that zircon can retain inherited signals under short events and thus motivating careful interpretation of single-grain outliers.

We also added the kinetic context that, under relatively short-duration high-temperature conditions, zircon fission-track annealing can be faster than He diffusion (Reiners, 2005), which is widely used as a diagnostic framework for transient reheating and partial resetting.

Finally, we now explicitly interpret our results in light of this framework: the tight Middle Pleistocene cluster is taken to reflect effective resetting for most grains, whereas isolated older grains are treated as incompletely degassed/inherited (see Response to Comment 2).

(see lines 73–89 in the revised manuscript)

Comment 2

The manuscript reports a Middle Pleistocene (~0.63 Ma) age and additional Late Miocene ages. It is unclear whether these reflect multiple combustion episodes, partial resetting, or sample mixing. A more detailed explanation is needed on how these age populations were identified, and what geological processes they represent. In particular, how can the Late Miocene event be reconciled with the much younger Pleistocene age - do they correspond to different tectonic stages, or to local reactivation and erosion events?

Response:

We agree and have clarified both (i) how the age populations are identified and (ii) what geological processes they represent.

In our dataset, three zircon grains yield a tightly clustered Middle Pleistocene population (0.56–0.70 Ma; weighted mean 0.63 ± 0.19 Ma). In contrast, two grains yield isolated older ages (8.7 ± 0.5 Ma and 87.7 ± 5.4 Ma) that are single-grain results and strongly separated from the cluster. We therefore interpret the older ages not as a combustion episode, but as inherited thermochronologic signals retained due to incomplete degassing/partial resetting during the younger combustion event (especially plausible under transient heating).

To reconcile the Late Miocene signal with the Middle Pleistocene combustion age, we explicitly frame them as recording different stages of the regional uplift–exhumation history: Late Miocene uplift and denudation is widely documented around the eastern Junggar margin (Bogda and Kelameili), and this earlier phase could be preserved in incompletely reset grains; continued uplift and erosion later culminated in Middle Pleistocene near-surface exposure of Jurassic coal seams, triggering the combustion event recorded by the clustered ~0.63 Ma ages.

We also revised the abstract to clearly state that the older ages are interpreted as inherited signals rather than a definitive separate combustion event.

(see lines 300–303、366-375 and 405-416 in the revised manuscript)

Comment 3

While CM rock formation is indeed associated with near-surface exposure, it may also depend on factors such as coal rank, oxygen supply, aridity, and fracture permeability. Therefore, linking combustion ages directly to tectonic uplift could oversimplify the process. The authors should discuss possible non-tectonic controls and clarify how they distinguished tectonic from environmental triggers.

Response:

We agree and have expanded the discussion of non-tectonic controls. In the revised manuscript we now explicitly describe external and environmental factors including potential ignition sources (e.g., lightning/wildfires), arid–semi-arid conditions favoring long-term exposure and oxidation, and the role of groundwater/overburden in suppressing combustion under humid conditions.

(see lines 340–357 in the revised manuscript)

At the same time, we clarify our logic for relating combustion ages to tectonics: uplift and denudation are treated as the necessary boundary condition that brings coal seams to shallow, oxygen-accessible conditions, while permeability/fracturing pathways (faults/joints) and climate modulate the probability and persistence of combustion. We therefore avoid claiming a one-to-one deterministic link (“age = uplift time”), and instead interpret CM ages as a near-surface exposure/oxidation threshold marker that is most reasonably coupled to uplift/erosion in this structural context.

(see lines 368–375 and 430-435 in the revised manuscript)

Comment 4

The regional tectonic evolution of the eastern Junggar belt should be described more comprehensively. How do the derived ages compare with other thermochronological or geomorphic constraints from the nearby Bogda Shan, Kelameili, or Karamay regions? Including a regional synthesis (e.g., with published AFT, AHe, or sedimentary record data) would enhance the geological significance of this study.

Response:

We agree and have strengthened the regional synthesis. In the revised manuscript, we added a more comprehensive comparison with published constraints from adjacent regions. For example, we now summarize published AFT-based uplift/denudation phases from the northern Bogda Mountains (e.g., 12–7 Ma and since ~11 Ma) and from the Kelameili Mountains (rapid uplift/denudation since ~20 Ma), and we emphasize that many basin-margin thermal history models indicate accelerated uplift since ~10 Ma.

(see lines 405–410 in the revised manuscript)

To present this clearly, we compiled these regional constraints together with our CM-rock ages in a synthesis figure (Fig. 6) to highlight where our Late Miocene inherited signal and Middle Pleistocene combustion cluster fit within the broader eastern Junggar tectonic evolution.

Response to Reviewer #2

We sincerely thank you for your constructive and detailed comments, which helped us substantially improve the clarity, geological context, and interpretation of our dataset. We have carefully revised the manuscript accordingly. Below we provide a point-by-point response, with all changes implemented in the revised version.

Comment 1

Abstract: Please add more details regarding the data and results.

Response:

We have revised the Abstract to include more dataset- and result-specific information, including (i) the types of data collected (coal petrography/proximate–ultimate analyses; petrography and XRD mineralogy of CM rocks; zircon (U–Th)/He dating), and (ii) the key numerical result (clustered Middle Pleistocene ZHe age, 0.63 ± 0.19 Ma) and the presence/interpretation of isolated older single-grain ages.

(see lines 24–34 in the revised manuscript)

Comment 2

Line 53: The abbreviation is not necessary.

Response:

We agree and have removed the unnecessary abbreviation at its first occurrence and/or replaced it with the full term where it is not needed for readability.

Comment 3

Line 91: please add one or two sentence to introduce the tectonic setting or the regional geology of the Junggar basin…

Response:

We added a concise introduction to the regional tectonic/geomorphic setting of the Junggar Basin and its peripheral orogenic belts in the Introduction, and we strengthened the linkage to the eastern Junggar structural framework in the Geological Setting. The study area is now explicitly placed within the eastern uplift of the Junggar Basin, bounded by the Kelameili Mountains to the north and the Bogda Mountains to the south, and within the Shazhang Fault–Fold Belt structural subdivision. Relevant references (including those suggested by the reviewer) have been incorporated.

(see lines 111、117-118 and 122 in the revised manuscript)

Comment 4

Line 101: “it”, please specify.

Response:

We revised the sentence by explicitly stating the antecedent of “it” ， “Shazhang Fault–Fold Belt”

(see line 117 in the revised manuscript)

Comment 5

Line 142: Which study area? Please specify clearly.

Response:

The study area is now explicitly defined as the Shazhang Fault–Fold Belt.

(see line 159 in the revised manuscript)

Comment 6

Line 198: Section 4.1.1; Line 206: Section 4.1.2; Line 215: Section 4.1.3.

Response:

Thank you for noting this. We corrected the internal cross-references and ensured they point to the correct subsections (4.1.1–4.1.3) in the revised manuscript.

(see lines 219、228 and 237 in the revised manuscript)

Comment 7

Section 4.3: “and another 87.7 ± 5.4 Ma; due” — how should this be interpreted? Why was this data point discarded?

Response:

We revised Section 4.3 and the Discussion to clearly explain the interpretation and treatment of the two isolated older single-grain ages (8.7 ± 0.5 Ma and 87.7 ± 5.4 Ma). Specifically:

Main age population: Three grains yield a tight cluster (0.56–0.70 Ma) with a weighted mean of 0.63 ± 0.19 Ma, interpreted as the timing of the combustion event that formed the CM rocks.

Isolated older ages: The 8.7 Ma and 87.7 Ma results occur as single-grain ages, strongly separated from the clustered Pleistocene population. We therefore do not include them in the weighted mean and treat them as inherited/partially reset signals, likely reflecting incomplete He loss during a short-lived heating event and/or inherited thermochronologic information.

We also clarified wording so the reader understands these data were not “discarded” as errors, but rather excluded from the mean age calculation because they are not part of the principal combustion-reset age population.

(see lines 301–303 、366-375 in the revised manuscript)

Comment 8

Section 5.1: Are these strata currently undergoing spontaneous combustion, or did combustion occur in the Cenozoic?… If combustion temperature exceeded the closure temperature… would that imply thermochronologic ages might not reflect uplift or exhumation?

Response:

We appreciate this important point and expanded Section 5.1 and the interpretation framework to avoid overstatement. Our revised text clarifies that:

The ZHe ages from CM rocks primarily date the combustion-related heating (and associated exposure to oxygen-rich near

---

## [Decision Letter · Decision Letter 1]

3 Mar 2026

PONE-D-25-53522R1Late Cenozoic exhumation in the eastern Junggar Basin: Evidence from zircon (U-Th)/He ages of combustion metamorphic rocksPLOS One

Dear Dr. Chen,

Thank you for submitting your manuscript to PLOS ONE. After careful consideration, we feel that it has merit but does not fully meet PLOS ONE’s publication criteria as it currently stands. Therefore, we invite you to submit a revised version of the manuscript that addresses the points raised during the review process.

We look forward to receiving your revised manuscript.

Kind regards,

Hu Li

Academic Editor

PLOS One

Journal Requirements:

Reviewers' comments:

Reviewer's Responses to Questions

**Comments to the Author**

1. If the authors have adequately addressed your comments raised in a previous round of review and you feel that this manuscript is now acceptable for publication, you may indicate that here to bypass the “Comments to the Author” section, enter your conflict of interest statement in the “Confidential to Editor” section, and submit your "Accept" recommendation.

Reviewer #1: All comments have been addressed

Reviewer #2: All comments have been addressed

2. Is the manuscript technically sound, and do the data support the conclusions?

Reviewer #1: Yes

Reviewer #2: Yes

3. Has the statistical analysis been performed appropriately and rigorously? 

Reviewer #1: Yes

Reviewer #2: I Don't Know

4. Have the authors made all data underlying the findings in their manuscript fully available?

Reviewer #1: Yes

Reviewer #2: Yes

5. Is the manuscript presented in an intelligible fashion and written in standard English?

Reviewer #1: Yes

Reviewer #2: Yes

6. Review Comments to the Author

Reviewer #1: I thank the authors for their careful and thorough revision of the manuscript. The revised version has addressed the major concerns raised in previous review in a satisfactory manner. In particular, the authors have improved the discussion of transient high-temperature heating and zircon (U-Th)/He resetting, clarified the interpretation of the age populations, expanded the treatment of non-tectonic controls on coal combustion, and strengthened the regional tectonic synthesis. Overall, the manuscript is significantly improved in clarity, balance, and geological context, and now presents a more cautious and well-supported interpretation of the dataset.

Minor Comments

1. Although the authors appropriately discuss partial resetting and inherited signals, the zircon (U-Th)/He dataset remains relatively small. It may be helpful to add a brief statement in the Discussion acknowledging the limited number of dated grains and noting that future studies with larger datasets would further test and refine the proposed interpretation.

2. The manuscript refers to long-lasting coal-fire events in some settings. Where possible, the authors may wish to clarify whether such durations are directly documented in the study area or inferred from analogs, in order to avoid potential overgeneralization.

3. Please ensure that Figs. 4 and 6 and related synthesis figures are sufficiently clear and readable at publication scale.

4. A careful proofreading is recommended to correct minor typographical errors and improve sentence flow in several places (e.g., in the Abstract and Section 5).

Reviewer #2: The authors revised the manuscript (#PONE-D-25-53522R1) according to the comments. The revised manuscript has been improved essentially. To improve the impact of the manuscript, here are some minor concerns.

Fig.1: fold-thrust belt; Anticline. Please double check throughout the manuscript.

Fig.6: Li et al., 2008

The ages of 8.7 Ma and 87.7 Ma are interpreted as “inherited zircon (U–Th)/He ages,” but no supporting evidence is provided. For example, no analyses of zircon morphology, inclusions, or U–Th distribution were conducted to exclude the effects of crystal heterogeneity. The possibility that these ages represent partially reset residual ages is not discussed, nor is it stated whether thermal history modeling (e.g., HeFTy or QTQt) was performed to test the possibility of multi-stage cooling. If possible, it is recommended to include correlations between single-grain zircon U–Th contents, eU values, and ages to evaluate whether the anomalous ages are related to radiation damage.

Zircon (U–Th)/He ages are commonly influenced by radiation damage, which typically results in a negative correlation between eU and age. However, the manuscript does not present an eU–age plot, making it difficult to assess whether the age dispersion is related to eU. If possible, the authors are encouraged to add an eU–age scatter plot and discuss whether differential He retention due to radiation damage accumulation could explain the observed age variability.

7. PLOS authors have the option to publish the peer review history of their article (what does this mean?). If published, this will include your full peer review and any attached files.

Reviewer #1: **Yes:**Zhiyuan He

Reviewer #2: **Yes:**Liang Qiu

---

## [Author Response · Author response to Decision Letter 2]

7 Apr 2026

Manuscript title: Late Cenozoic exhumation in the eastern Junggar Basin: Evidence from zircon (U-Th)/He ages of combustion metamorphic rocks

Manuscript ID: PONE-D-25-53522R1

We sincerely thank the Academic Editor and the reviewers for their careful evaluation of our manuscript and for the constructive suggestions that helped us improve the paper. We have revised the manuscript accordingly and carefully polished the language throughout. Our detailed responses are provided below.

Reviewer #1

1. Although the authors appropriately discuss partial resetting and inherited signals, the zircon (U-Th)/He dataset remains relatively small. It may be helpful to add a brief statement in the Discussion acknowledging the limited number of dated grains and noting that future studies with larger datasets would further test and refine the proposed interpretation.

Response: We agree and have added an explicit limitation statement in Section 5.2. The revised text now notes that only five usable single-grain ages were obtained from a single sample, that the present dataset is not sufficient for robust inverse thermal-history modelling, and that future work using larger single-grain datasets together with zircon imaging and complementary thermochronometers will be necessary to refine the interpretation of the anomalously old grains.

(see lines 377-385, 415-451 in the revised manuscript)

2. The manuscript refers to long-lasting coal-fire events in some settings. Where possible, the authors may wish to clarify whether such durations are directly documented in the study area or inferred from analogs, in order to avoid potential overgeneralization.

Response: We agree and have revised the relevant sentence in the Introduction. In the original manuscript, the statement that coal-fire-related heating may persist for hundreds of years was based on published studies from other regions, particularly the Powder River Basin, rather than on direct chronological or observational constraints from the eastern Junggar Basin. We have therefore revised the text to make this distinction explicit. In the revised manuscript, we now clarify that prolonged coal-fire durations are documented in analogous natural coal-fire systems elsewhere, whereas in our study area the available evidence primarily constrains the timing of combustion metamorphic rock formation rather than the exact duration of individual combustion events. This revision avoids overgeneralization and better reflects the limits of the present dataset.

(see lines 73-77 in the revised manuscript)

3. Please ensure that Figs. 4 and 6 and related synthesis figures are sufficiently clear and readable at publication scale.

Response: We have checked the figure captions and manuscript references carefully with a simplified layout suitable for publication scale.

(see lines 282, 431 in the revised manuscript)

4. A careful proofreading is recommended to correct minor typographical errors and improve sentence flow in several places (e.g., in the Abstract and Section 5).

Response: We have carefully proofread the entire manuscript and revised the Abstract and Section 5 for clarity, concision, and sentence flow. Minor typographical issues, capitalization inconsistencies, and wording problems were corrected throughout.

(see lines 14-32 and 360-388 in the revised manuscript)

Reviewer #2

1. Fig.1: fold-thrust belt; Anticline. Please double check throughout the manuscript.

Response: We have double-checked the terminology throughout the manuscript and standardized capitalization and usage of structural names. In particular, occurrences such as 'Huoshaoshan anticline' and 'Xidagou syncline' were revised to 'Huoshaoshan Anticline' and 'Xidagou Syncline' where they refer to formal structural units. In the manuscript text, we have also standardized the terminology by changing 'Fault-Fold Belt' to the proper format 'Fold-Thrust Belt'.

2. The ages of 8.7 Ma and 87.7 Ma are interpreted as 'inherited zircon (U-Th)/He ages,' but no supporting evidence is provided. For example, no analyses of zircon morphology, inclusions, or U-Th distribution were conducted to exclude the effects of crystal heterogeneity. The possibility that these ages represent partially reset residual ages is not discussed, nor is it stated whether thermal history modeling (e.g., HeFTy or QTQt) was performed to test the possibility of multi-stage cooling. If possible, it is recommended to include correlations between single-grain zircon U-Th contents, eU values, and ages to evaluate whether the anomalous ages are related to radiation damage.

Response: We appreciate this important suggestion and have revised both the Methods and Discussion accordingly. First, we now state explicitly in Section 3.7 that the dated zircons were clear, euhedral, inclusion-free under binocular and polarized-light inspection, and that all dated grains are euhedral zircons with two terminations. Second, we now clarify that no cathodoluminescence imaging, internal U-Th zoning analyses, or inverse thermal-history modelling (e.g., HeFTy/QTQt) were performed, and we explain that the available dataset is too limited to support robust thermal-history inversion. Third, we revised the interpretation of the older grains to be more cautious: rather than treating them unequivocally as inherited ages, we now interpret them as incompletely reset or inherited pre-combustion thermochronologic components. Finally, we added an eU-age comparison and discuss its implications in the Results and Discussion.

(see lines371-385, 415-451 in the revised manuscript)

3. Zircon (U-Th)/He ages are commonly influenced by radiation damage, which typically results in a negative correlation between eU and age. However, the manuscript does not present an eU-age plot, making it difficult to assess whether the age dispersion is related to eU. If possible, the authors are encouraged to add an eU-age scatter plot and discuss whether differential He retention due to radiation damage accumulation could explain the observed age variability.

Response: We agree and have expanded this point in the revised manuscript. We further compared the single-grain zircon (U-Th)/He ages with the U and Th contents reported in Table 2. These comparisons show similarly non-systematic patterns: the 87.7 Ma grain does not have the highest U or Th content, the 8.7 Ma grain has the highest eU value, and the three Middle Pleistocene grains remain tightly clustered in age despite substantial variation in U, Th, and eU. We therefore interpret the age dispersion cautiously and conclude that radiation damage alone is insufficient to explain the observed spread. Instead, the two anomalously older grains are more plausibly interpreted as incompletely reset or inherited pre-combustion components. We also now state explicitly that, because only five usable grains were obtained from a single sample, these age-composition relationships should be regarded as exploratory rather than statistically robust.

(see lines 366-376 in the revised manuscript)

We again thank the reviewers for their constructive comments, which have substantially improved the manuscript.

---

## [Editor Report · Decision Letter 2]

13 Apr 2026

Late Cenozoic exhumation in the eastern Junggar Basin: Evidence from zircon (U-Th)/He ages of combustion metamorphic rocks

PONE-D-25-53522R2

Dear Dr. Chen,

We’re pleased to inform you that your manuscript has been judged scientifically suitable for publication and will be formally accepted for publication once it meets all outstanding technical requirements.

Kind regards,

Hu Li

Academic Editor

PLOS One
---

## [Editor Report · Acceptance letter]

PONE-D-25-53522R2

PLOS One

Dear Dr. Chen,

I'm pleased to inform you that your manuscript has been deemed suitable for publication in PLOS One. Congratulations! Your manuscript is now being handed over to our production team.

Kind regards,

on behalf of

Pro.Dr. Hu Li

Academic Editor

PLOS One